# An Integrated Approach Reveals DNA Damage and Proteotoxic Stress as Main Effects of Proton Radiation in *S. cerevisiae*

**DOI:** 10.3390/ijms23105493

**Published:** 2022-05-14

**Authors:** Laura Vanderwaeren, Rüveyda Dok, Karin Voordeckers, Laura Vandemaele, Kevin J. Verstrepen, Sandra Nuyts

**Affiliations:** 1Laboratory of Experimental Radiotherapy, Department of Oncology, KU Leuven, 3000 Leuven, Belgium; laura.vanderwaeren@kuleuven.be (L.V.); ruveyda.dok@kuleuven.be (R.D.); laura.vandemaele@hotmail.be (L.V.); 2Laboratory of Genetics and Genomics, Centre for Microbial and Plant Genetics, KU Leuven, 3000 Leuven, Belgium; karin.voordeckers@kuleuven.be; 3Laboratory for Systems Biology, VIB-KU Leuven Center for Microbiology, 3000 Leuven, Belgium; 4Department of Radiation Oncology, Leuven Cancer Institute, University Hospitals Leuven, 3000 Leuven, Belgium

**Keywords:** DNA damage response, proteotoxic stress, proton radiation, radiobiology, radiotherapy

## Abstract

Proton radiotherapy (PRT) has the potential to reduce the normal tissue toxicity associated with conventional photon-based radiotherapy (X-ray therapy, XRT) because the active dose can be more directly targeted to a tumor. Although this dosimetric advantage of PRT is well known, the molecular mechanisms affected by PRT remain largely elusive. Here, we combined the molecular toolbox of the eukaryotic model Saccharomyces cerevisiae with a systems biology approach to investigate the physiological effects of PRT compared to XRT. Our data show that the DNA damage response and protein stress response are the major molecular mechanisms activated after both PRT and XRT. However, RNA-Seq revealed that PRT treatment evoked a stronger activation of genes involved in the response to proteotoxic stress, highlighting the molecular differences between PRT and XRT. Moreover, inhibition of the proteasome resulted in decreased survival in combination with PRT compared to XRT, not only further confirming that protons induced a stronger proteotoxic stress response, but also hinting at the potential of using proteasome inhibitors in combination with proton radiotherapy in clinical settings.

## 1. Introduction

Today, up to 50% of cancer patients are treated with radiotherapy during their course of illness [1,2]. Radiotherapy strives for the optimal ratio between the dose delivered to the tumor and the lowest possible dose delivered to the surrounding healthy tissue. Current clinical practices mainly rely on classical X-rays or photon radiotherapy (XRT). However, because of the high entrance and exit dose of photons, damage to the healthy tissue surrounding the tumor remains a major problem associated with XRT [3].

Proton beam radiation (PRT) is an alternative form of radiation therapy that can be used to reduce the problem of the off-target toxicity associated with conventional XRT [4]. Due to their positive electrical charge and mass, protons are slowed down while penetrating tissues, and deposit most of their energy at the end of their path. These physical characteristics result in a unique, focused depth–dose distribution, the Bragg peak, allowing to spare the healthy tissues distal to the tumor from the toxic effects of the irradiation. The importance of PRT is well-recognized for the treatment of tumors close to critical structures; such as head-and-neck cancers; and is a treatment option for several cancers [5]. However, not all cancer patients benefit from the above described dosimetric advantage of protons, highlighting the need for understanding the underlying molecular mechanisms affected by PRT [6,7].

Studies comparing the cellular response to PRT and XRT on the level of survival, DNA damage induction or repair mechanisms affected, do not paint a clear picture [8,9,10,11,12,13,14,15,16,17]. One of the main reasons is the large variation in tumor biology which increases the complexity of interpreting these results. This hiatus in our understanding has limited the ability to achieve the true potential of PRT but also impeded the development of proton specific therapeutic and combinatorial strategies. Increased knowledge of the molecular pathways affected by PRT can lead to selective targeted therapy combinations. This is relevant in the field of personalized medicine because understanding the radiobiology of protons and its interaction with the complex biology of the tumor will allow for the integration of clinical, physical and biological parameters to adjust treatment to the specific needs of an individual patient. A system-wide approach, which is often missing, could help bring more insights into the molecular mechanisms affected by treatment with PRT.

Therefore, in this study, for the first time, we apply an integrated approach to identify the exact mechanisms and cellular pathways affected by both PRT and XRT using the unicellular eukaryote *Saccharomyces cerevisiae* as a model. The short life cycle, simple culture conditions and ease with which the genome can be manipulated make *S. cerevisiae* an extensively used model organism, not only in radiobiology but in medicine in general [18,19,20,21,22,23]. Moreover, the techniques and types of experiments available in this yeast make this model organism ideal for a systems biology approach, which is difficult to achieve in mammalian tumor cells because of the large heterogeneity between cell lines. Using Bar-Seq in combination with more detailed molecular assays, we found that genes involved in DNA repair determined the survival of cells exposed to XRT and PRT. Moreover, the DNA damage response was equally important for both irradiation types. Remarkably, in contrast to XRT, transcriptomic analysis after PRT showed a much stronger activation of genes involved in the response to proteotoxic stress. Additionally, inhibition of the proteasome resulted in decreased survival after PRT, but not after XRT. Altogether, our results offer a genome-wide view on the physiological effect of PRT and XRT and bridge the gap between current biological uncertainties and the translation of PRT to the clinic.

## 2. Results

### 2.1. Survival of S. cerevisiae Is Independent of the Radiation Type

*S. cerevisiae* has previously been used as a model to study the effects of radiation on cells, and is known to tolerate high doses of radiation [24,25,26,27,28,29,30,31,32]. Our experiments confirmed this tolerance to both PRT and XRT, with cells showing survival fractions of 0.44 ± 0.014 and 0.47 ± 0.054 after receiving a dose of 50 Gy PRT or XRT, respectively (Figure 1). Increasing the dose to 100 Gy resulted in only a minor decrease in the survival (0.41 ± 0.04 for PRT and 0.45 ± 0.023 for XRT), further suggesting a largely similar survival in response to PRT and XRT. Strikingly, although non-significant, the survival fractions obtained for PRT are systematically lower compared to XRT (except at a dose of 75 Gy) with PRT resulting in, on average, 1.05 ± 0.07 more cell kill.

### 2.2. Transcriptomic Response in S. cerevisiae Is Dependent on Radiation Type

The largely similar survival rates of *S. cerevisiae* in response to PRT or XRT do not exclude possible differences in the underlying physiological response to both types of radiation [33,34]. Therefore, we used RNA sequencing (RNA-Seq) to assess and compare the transcriptional response to a 50 Gy PRT and XRT treatment. As previously determined, this dose resulted in an average survival fraction of approximately 45%. The transcriptome (~6600 genes) was assessed at an early (30 min after radiation) and later (90 min after radiation) timepoint.

Principal component analysis (PCA) of the transcriptome data for the different treatments showed distinct clustering of the proton and photon samples and the respective non-irradiated controls (Appendix A). PRT and XRT experiments were analyzed separately by calculating the Log2 fold changes (log2FC) of the irradiated condition compared to the corresponding control. To select significantly up- or downregulated genes, the false discovery rate (FDR) was set at 0.01 and a cut-off log2FC of 1.5 for upregulated and −1.5 for downregulated genes was used. Using these cut-offs, volcano plots showed that 68 and 104 genes were differentially expressed compared to cells that did not receive a treatment at 30 and 90 min after PRT, respectively. For XRT, only 37 genes showed differential expression patterns at 30 min, and this further decreased to only 24 genes at 90 min (Appendix A), suggesting that the transcriptional response to PRT is different and stronger compared to the response to XRT. In line with these data, hierarchical clustering of the sets of differentially regulated genes showed that the samples were clustered per irradiation type (Figure 2A).

Gene Set Enrichment Analysis (GSEA) showed that deoxyribonucleotide biosynthetic process is highly enriched among the upregulated genes at both timepoints after irradiation and independent of the radiation type (Figure 2A,B). The upregulated genes in this gene ontology (GO) term are primarily involved in DNA metabolism. Among these genes are four genes that encode the subunits of the ribonucleotide reductase (RNR) complex which catalyzes the rate-limiting step in deoxyribonucleoside triphosphates (dNTP) synthesis: *RNR1*, *RNR2*, *RNR3* and *RNR4*. dNTP levels are known to increase after DNA damage and are reported to indicate active DNA repair [35].

Although less pronounced, a second group of genes that were significantly upregulated, independent of radiation type, are involved in the cellular response to DNA damage stimulus (Figure 2A,B). This includes the genes encoding the cell cycle checkpoint kinase Dun1, the DNA glycosylase Mag1 that initiates the base excision repair pathway, and Rad51, which functions in strand exchange during homologous recombination. Ninety minutes after radiotherapy, this upregulation was less pronounced, which corresponds to the GSEA analysis (Figure 2B). PRT, but not XRT, also caused a slight induction of genes involved in glycolysis. This includes the genes encoding glucokinase *GLX1*, pyruvate kinase *PYK2*, hexokinase *HXK1* and glyceraldehyde-3-phosphate dehydrogenase *TDH1*. It has previously been reported that yeast metabolism can be affected by various types of radiation treatment, but can also reflect the environmental stress response of yeast activated after various kinds of stresses [12,36,37,38].

Interestingly, a large set of genes was only or much stronger expressed after PRT, among which the genes *SIS1*, *STI1*, *CUR1*, *SSA1*, *SSA2*, *BTN2*, *HSP42*, *HSP78*, *HSP82* and *HSP104* which are involved in protein folding (Figure 2A). The upregulation of these genes was even stronger 90 min after irradiation. Indeed, GSEA revealed that genes that fall under the GO classes of “protein refolding” and “protein targeting to the mitochondria” were significantly enriched 90 min after PRT (Figure 2B). These GO classes encompass the chaperones *HSP78*, *HSP82*, *HSP42*, *SSA1* and *SSA2* and the disaggregase *HSP104.* These genes were upregulated with induction levels ranging from of 1.6 to 3.5 log2 fold changes, suggesting that some form of proteotoxic stress occurs after PRT. Additionally, XRT induced some genes involved in protein folding namely *CUR1*, *SSA1* and *HSP42* 30 min after radiation. Additionally, 90 min after irradiation, *BTN2*, *HSP42*, *HSP78*, *HSP82* and *HSP104* were induced. However, the changes in expression observed after XRT were less strong compared to those elicited by PRT, with log2 fold changes after XRT ranging from 0.6 to 1.3.

A differential response was observed for the downregulated genes comparing the response 30 and 90 min after PRT and XRT. Hierarchical clustering for the downregulated genes showed four main groups (Figure 2A). The first group was enriched in genes categorized under the GO class “chromatin assembly or disassembly” (Figure 2B). This group is mainly composed of the histone-encoding genes *HTA1*, *HTA2*, *HTB2*, *HHF1* and *HHF2* that were highly downregulated at 30 min after PRT and XRT, and less strongly at 90 min after PRT and XRT. A second group of downregulated genes was involved in cell separation after mitosis and corresponds to the GO term “septum digestion after cytokinesis”, which was significantly enriched at 30 min after XRT. This group is composed of the genes *CTS1*, *DSE1*, *DSE4*, *EGT2* and *SDS24*. For samples treated with PRT, two more significant GSEA enriched sets were identified, namely “transfer RNA gene-mediated silencing” and “DNA repair”. Both GO classes encompass the histone encoding genes *HTA1*, *HTA2* and *HHF1*. Overall, these patterns of gene downregulation are in-line with reduced cell division. Histones are known to be coupled with DNA synthesis, and downregulation of histones is an indication of cell cycle checkpoint activation [39,40]. This is commonly seen after radiotherapy: when cells are coping with DNA damage, DNA damage checkpoints are activated and the progression through the cell cycle is slowed down to allow sufficient time for DNA repair.

Network analysis of the differentially expressed genes showed three distinct clusters (Figure 2C). Genes facilitating DNA repair, namely the genes involved in deoxyribonucleotide biosynthetic process, cellular response to DNA damage and chromatin assembly or disassembly cluster together. This part of the network consists of genes deregulated after both PRT and XRT. Genes involved in yeast metabolism, and the general stress response forms one big cluster together with the heat shock proteins. A separate cluster is formed by the genes categorized under the GO term “septum digestion after cytokinesis” that were deregulated after XRT. Network analysis also highlighted that although a large set of genes displayed changed expression levels after both PRT and XRT treatment (37 out of 139 genes), PRT caused deregulation of many more genes. An additional set of 94 genes were specifically deregulated after PRT compared to only 8 after XRT.

### 2.3. DNA Repair Determines the Survival of S. cerevisiae after PRT and XRT

RNA-Seq analysis strongly suggests that both PRT and XRT evoke a similar DNA damage response. However, the response after PRT seems slightly more complex, with the stronger induction of genes involved in proteotoxic stress. To further investigate these potential differences in molecular mechanisms affected after PRT and XRT, we performed a barcode sequencing (Bar-Seq) experiment to assess which genes are important to ensure survival after PRT or XRT. Using the haploid yeast deletion collection, we tested the complete collection of approximately 4800 mutants, each carrying a deletion for one of the 4800 non-essential *S. cerevisiae* genes. Importantly, each mutant also carries unique DNA barcodes, allowing to assess their sensitivity to PRT and XRT in a high-throughput assay termed “Bar-Seq” (for more details, see materials and methods and [41,42]). The deletion mutant pools were irradiated with a dose of 50 Gy PRT or XRT, while a control population received a mock treatment (i.e., undergoing a similar experimental procedure, but without actual radiation).

The results of the Bar-Seq experiment revealed 20 and 35 genes that, when deleted, caused sensitivity to PRT and XRT, respectively (FDR 0.05 and log2FC < −0.5). The most sensitive deletion strains were depleted with a log2FC of −3.42 and −3.51 relative to the control for the PRT and XRT, respectively. Similar as for the RNA-Seq experiment, multidimensional scaling (MDS) plots showed clustering according to the radiation type and within their respective controls (Appendix A). Figure 3A shows all the genes for which the cut-offs are met in either or both the PRT and XRT experiment. GSEA revealed an enrichment of DNA repair pathways for both PRT and XRT (Figure 3B). Moreover, the top five most sensitive deletion mutants were shared between PRT and XRT, namely deletions for the genes *RAD50*, *RAD51*, *RAD52*, *RAD55* and *XRS2*. For the genes that are important for either PRT or XRT, no GO enrichment could be found. Network analysis showed a large cluster of genes involved in DNA repair shared between the PRT and XRT experiment (Figure 3C).

The sensitivity of these genes for PRT and XRT was confirmed by plating assays (Appendix A). Genes not involved in DNA repair also did not cluster in the network confirming the lack of a functional enrichment for these genes.

Bar-Seq also identified a few genes that when deleted reduced sensitivity to PRT or XRT. However, no functional enrichment for these genes could be identified. Moreover, these genes were often not shared between the PRT and XRT experiment. Additionally, the reduced sensitivity could not be validated by plating assays. Therefore, we only focused on the genes that, when deleted, cause sensitivity to PRT or XRT.

Constructing an interaction network for all genes found in either or both RNA-Seq and Bar-Seq highlighted that both experiments detected the importance of the DNA damage response after PRT or XRT; the experiments overlapped at the level of the GO term “response to DNA damage stimulus” (Appendix A). Two genes, *RAD51* and *DUN1*, both involved in the DNA damage response, showed a differential expression in the RNA-Seq at 30 min after PRT and sensitivity in the Bar-Seq in after PRT. However, this does not imply that the other genes are less important, since not all key genes needed for DNA damage survival are induced by the stress [45]. Even though heat shock proteins were upregulated in the response to PRT and XRT, the Bar-Seq experiment revealed that none of the proteins by themselves are essential for survival following PRT or XRT. This is likely due to the redundancy among heat shock proteins [46,47,48,49]; where gene products are often functionally similar.

### 2.4. PRT Induces DNA Damage and Protein Damage in S. cerevisiae

Our results so far confirmed the importance of DNA repair in response to XRT, and showed that the same response is also crucial for surviving PRT. Next, we set out to further investigate whether both radiation types induced a similar amount of DNA damage. First, we investigated the kinetics of double strand break repair by tracking cellular Rad52 foci under the microscope after radiotherapy, which serve as a marker for double strand breaks (Figure 4A) [50]. PRT and XRT both showed an induction of Rad52 foci with a maximum fold increase of 5.8 and 6.2 both at 120 min after 50 Gy PRT and XRT, respectively. A dose of 100 Gy resulted in a maximal 6.1-fold increase in foci 120 min after PRT, and a maximal increase of 6.3-fold 90 min after XRT (Figure 4A). Although the increase in Rad52 foci was comparable between the two radiation types, the dynamics appear slightly different. More Rad52 foci were observed at 30 min after PRT compared to XRT. PRT caused 4.33- and 5.88-fold induction after 50 Gy and 100 Gy, respectively, while XRT-induced Rad52 foci increased 4.19- and 4.03-fold after 50 Gy and 100 Gy, respectively. Moreover, after 150 min, more Rad52 foci have been cleared in cells subjected to XRT compared to PRT (23% versus 28% at 50 Gy and 25% versus 30% at 100 Gy after XRT and PRT, respectively), indicating PRT-induced DNA damage is more persistent.

Next, we investigated possible changes in cell cycle dynamics after irradiation. Cell cycle arrest is generally recognized as an indispensable part of the DNA damage response. For example, previous studies showed that Gamma-irradiated synchronized yeast cultures show a prolonged G1 arrest that is dose-dependent and proportional to the amount of DNA damage [51]. In line with this previous result, we found that irradiation induced a dose-dependent increase in cells that are in the G1 phase, independent of the irradiation type (Figure 4B). The G1 arrest became less pronounced after 90 min for both PRT and XRT after 50 Gy and 100 Gy, although the 100 Gy treatment resulted in slightly more cells remaining in G1.

Besides the importance of the DNA damage response, RNA-Seq analysis revealed that genes involved in protein folding, mostly heat shock proteins, were more highly upregulated after PRT. To further investigate this, we examined the amount of protein aggregates after PRT and XRT by fluorescently tagging *HSP104*, a disaggregase that localizes to protein aggregates [52,53,54,55]. As *S. cerevisiae* is radioresistant, we expected protein aggregates to be cleared efficiently by the proteostasis machinery. Therefore, we additionally inhibited the central degradation machinery of protein control, the proteasome, using the potent proteasomal inhibitor MG-132. This is commonly performed to study protein aggregate formation [55,56]. Both PRT and XRT showed an increase in Hsp104 foci with a maximum of 29% foci positive cells at 120 min after 50 Gy PRT and 35% at 90 min after 50 Gy XRT (Figure 4C), indicating the presence of protein aggregates after both radiation types. This further suggests the induction of proteotoxic stress after both PRT and XRT, confirming the results from our RNA-Seq experiment. At 50 Gy the maximum amount of Hsp104 foci is reached at 90 min after XRT versus 120 min after PRT, indicating a delay of protein aggregate formation after PRT. Compared to the maximum, at 150 min after 50 Gy PRT the amount of Hsp104 foci has decreased from 29% to 28% while after XRT the decrease was more pronounced (from 35% to 31%). At 100 Gy a different profile was observed: while PRT-induced Hsp104 foci were still increasing from 90 to 150 min (from 18% to 23% to 26%), XRT-induced Hsp104 foci remained more constant (26% to 29% to 27%). As a result, similar to the kinetics observed for DNA damage markers, PRT-induced Hsp104 foci, indicative for protein aggregates, showed a more persistent pattern compared to XRT. We next investigated if differences in the aggregates formation could be linked to an increase in endoplasmic reticulum (ER) stress, since ER stress is one of the consequences of increased unfolded or misfolded proteins [57,58]. Moreover, transcriptome analysis revealed that genes involved in protein refolding were upregulated after 50 Gy PRT, strengthening the hypothesis of the presence of unfolded or misfolded proteins after PRT. We therefore measured the activation of the unfolded protein response (UPR), which is known to be a good marker for ER stress [59,60]. Specifically, we assayed the *HAC1* mRNA splicing status by PCR across the intron of *HAC1* mRNA [58,61,62]. The results showed activation of the UPR for both PRT and XRT, confirming that both PRT and XRT induce proteotoxic stress, although the extent of UPR activation varied between replicates at different timepoints (Appendix A). Next, to assess the effect of the radiation-induced protein damage on survival of *S. cerevisiae*, we performed spotting assays with the proteasomal inhibitor MG-132 in combination with RT. In line with the presence of more persistent protein aggregates, PRT resulted in stronger growth inhibition compared to XRT upon proteasome inhibition with MG-132 (Figure 4D).

## 3. Discussion

To gain more insight into the molecular mechanisms affected by PRT, we used *S. cerevisiae* as a model to investigate PRT radiobiology on a functional and transcriptional level and compared it with conventional XRT. To our knowledge, this is the first time an integrated systems biology approach has been used to shed light on the molecular mechanisms affected by PRT.

The ability to repair DNA damage caused by ionizing radiation has been shown to be a central factor in the response to XRT in several different organisms [26,63,64]. This agrees with our observations, which show an induction of genes involved in the DNA damage response following XRT. Moreover, our functional assays also showed that deleting certain central DNA damage response genes severely affects survival after XRT. Importantly, we find comparable results after PRT, suggesting that for both types of radiation, DNA repair genes are important for survival. Multiple studies in mammalian cells suggest that PRT-induced DNA lesions are more dependent on homologous recombination for repair, and less on the more common mechanism of non-homologous end joining [16,17,33,65]. Our results confirmed the involvement of homologous recombination for survival after PRT. However, it has to be noted that the pathway for non-homologous end joining is much less used by yeast cells [66,67], which implies that we cannot exclude that this pathway is also involved in repair after PRT in other organisms. Furthermore, the Bar-Seq experiment also identified several genes that affect survival but are not involved in DNA repair. However, we were unable to confirm their role through plating assays. This could be because, as any genome-wide screen, Bar-Seq often yields a few false positive hits, or because the plating assays are less suitable to detect subtle differences in survival after RT.

The importance of the DNA damage response after PRT and XRT was further examined by assessing the kinetics of the DNA damage response through tracking of Rad52 foci and characterizing cell cycle kinetics. Both PRT and XRT resulted in similar induction of DNA damage and G1 cell cycle phase arrest. However, clearance of DNA damage seemed slightly slower after PRT, indicating possible differences in the kinetics of DNA damage or repair. In mammalian cell research, contradictory results have been reported, with some studies detecting a prolonged cell cycle arrest after PRT [68,69,70], while others found no difference [13], or report a shorter cell cycle arrest for PRT [16].

Apart from DNA damage, radiation is known to also induce protein damage, which could make a substantial contribution to the radiobiological response that is elicited [71,72]. Strikingly, our transcriptional analysis reveals that PRT elicits a much stronger induction of genes involved in proteotoxic stress, for example genes encoding chaperones. Heat shock proteins play a significant role in the protein folding and the proteostasis machinery in general. However, they also help maintain efficient DNA repair, by stabilizing specific protein complexes (reviewed in Dubrez et al., 2020) [73]. Here, using proteasomal inhibition, we demonstrated a comparable induction of protein aggregates after PRT and XRT. Like the DNA damage response kinetics, we noted a slightly more persistent response for PRT, suggesting subtle differences in the induction and/or clearance of protein damage. Importantly, proteasomal inhibition in combination with PRT resulted in more extensive growth inhibition compared to XRT, suggesting that PRT elicits a stronger proteotoxic stress, resulting in a greater dependence on proper proteasomal functioning after PRT. It is interesting to note that in the study of Schultzhaus et al. the GO term “protein catabolism” was also slightly more upregulated in the particle irradiation group including PRT, deuterons and α-particles, compared to gamma-irradiation. However, in contrast to our study, the potential role and kinetic of this process was not further investigated.

Despite the stronger transcriptional and molecular response after PRT, our results show that irradiation of *S. cerevisiae* with medium to high doses (25 to 100 Gy) of PRT and XRT resulted in similar survival curves. However, it should be noted that, although non-significant, PRT resulted in on average 1.05 ± 0.07 times more cell killing compared to XRT.

We believe that our study offers more robust conclusions than previous work because the use of the eukaryotic model *S. cerevisiae* allowed for us to perform multiple replicates of very controlled genome-scale analyses, which allowed us to gain a complete view on the cellular response to PRT and XRT. In addition, we were able to verify the results of our genome-wide assays, RNA-Seq and Bar-Seq, by probing the consequences of deleting individual genes and by performing specific experiments looking in more detail at the affected molecular mechanisms. However, it should be underlined that our study also has some limitations. Along with the existing molecular toolbox, the large body of knowledge on its genetics and physiology, and the conserved nature of core processes such as the DNA damage response, caused *S. cerevisiae* to be a particularly suitable and popular model to study eukaryotic radiobiology [23,24,25,26,31,32]. However, as is the case for all studies using models, the results cannot be expanded to other, more complex, organisms without verification. In addition, a quantitative comparison between XRT and PRT is difficult because the treatments are carried out with different irradiation machines stationed at distinct locations, which implies the risk of minor differences that are independent of the radiation type. It also is important to note that PRT assays were performed in the plateau region of the depth–dose profile, i.e., at a depth located before the Bragg peak, to ensure that all cells would receive a uniform dose. This plateau region is characterized by a lower energy deposition and hence lower cell toxicity compared to the Bragg peak region, which could explain why we did not observe a statistical difference in biological effectiveness between PRT and XRT.

Thus, while the results clearly show a trend of similarities in the response to PRT and XRT, the more subtle differences we find in DNA and protein damage must further be investigated. However, the increased dependency of proper proteasomal functioning after PRT, in particular, offers novel therapeutic potential for proteasome inhibitors in combination with PRT. While the results need to be confirmed in higher organisms, our genome-scale analysis is difficult to achieve in mammalian tumor cells because the large heterogeneity between cell lines and the more restricted experimental possibilities. By offering a first in-depth view on the radiobiological response of *S. cerevisiae* to PRT in comparison to the more conventional XRT, our study provides a good basis for further, more targeted analyses and thus helps bridge the gap between the current biological uncertainties and translation of PRT to clinical settings.

## 4. Materials and Methods

### 4.1. Strains and Media

All strains used in this study were derived from the haploid auxotrophic S288c strain BY4741 (MATa). A full list of strains used can be found in Appendix A. Primers used for strain construction and verification were ordered from IDT. Primer sequences are listed in Appendix A.

Deletion strains were constructed with a HygB cassette amplified from the pCB1 plasmid using primers with at least 40 bp sequence homology to the target DNA [74]. For fluorescent tagging, yECitrine was amplified from the pKT140 plasmid using primers with 40 bp sequence homology to the target DNA [75]. A list of plasmids used in this study and their genotype can be found in Appendix A. PCR products were used for yeast transformation using a LiAc-based procedure. Transformants were verified by PCR using specific check primers.

Yeast was cultured in peptone dextrose (YPD) medium containing 2% bacteriological peptone (Lab M, Heywood, UK), 1% yeast extract (Lab M, Heywood, UK) and 2% glucose (Sigma, St. Louis, MO, USA). For solid plates 2% agar (Lab M, Heywood, UK) was added. YPD containing 200 µg/L Hygromycin B (Invitrogen, Waltham, MA, USA) or 200 µg/mL G418 (Formedium, Swaffham, UK) was used for selection of yeast transformants.

### 4.2. Irradiations

Photon irradiations (6 MV, 5.3 Gy/min, LET = 2.36 keV/µm [76]) were performed on a TrueBeam medical linear accelerator (Varian Medical Systems, Palo Alto, CA, USA) placed at the Department of Radiation Oncology in University Hospitals Leuven. Proton irradiation (62 MeV, 17 Gy/min, mean LET = 10.51 keV/µm) was carried out at the Centre-de-Ressources du Cyclotron (Université Catholique de Louvain, Louvain-la-Neuve, Belgium). Samples were irradiated at the entrance of the proton depth–dose curve. Due to the separate locations of the two irradiation facilities, photon and proton experiments were never performed on the same day. For both irradiation types a physical dose of 50 Gy or 100 Gy was used. The biological effects of the physical radiation dose for both irradiation types were determined by investigation of relaxation patterns of 90% supercoiled phiX174 RF1 DNA (0.5 g/L DNA, Thermo Fisher Scientific, Waltham, MA, USA) as previously described [77,78]. The 5386 bp ϕDNA was diluted to a final concentration of 7.5 nM and mock-irradiated or irradiated with 5 and 15 Gy of protons and photons. Irradiated samples were analyzed on a 1% agarose gel. Densitometry was performed in ImageJ (Appendix A). The percentage of DNA damage, which was calculated as the ratio of circular DNA (cDNA) and linear DNA (lDNA) over the total amount of DNA (scDNA + cDNA + lDNA), showed a dose-dependent increase independent of the radiation type.

### 4.3. Plating Assays

Plating assays were performed in triplicate to determine the survival of yeast after PRT or XRT. Cells in exponential phase (OD ~0.2) were irradiated with indicated doses of PRT or XRT and were plated on YPD. The number of colonies was determined after 48 h of growth at 30 °C. Plates were considered countable when containing 30–300 colonies. The survival fraction was calculated by dividing the number of colonies from the irradiated conditions by the control condition.

### 4.4. RNA Sequencing and Analysis

Four replicate exponential phase cultures of the wild type strain (KV447) were irradiated with 50 Gy PRT or XRT. At 30 min and 90 min after irradiation samples were taken for RNA extraction. Therefore, 1.5 mL of the culture was centrifuged and after washing the pellet once in ice-cold DEPC water, the pellet was put on dry ice. Pellets were stored at −80 °C until RNA extraction. RNA was extracted using the MasterPure Yeast RNA Purification Kit (Lucigen, Middleton, WI, USA) with DNase treatment. RNA concentrations were measured with Qubit (Thermofisher). RNA sequencing was performed on a DNBseq platform at BGI Genomics. Sequences were assessed for quality control using FASTQC version 0.11.9 and were aligned against the S288c genome of *S. cerevisiae* using HISAT2 version 2.2.1 [79,80]. Mapped reads were counted using HTSeq version 0.13.5 [81]. Differential expression analysis was performed in edgeR 3.32.1 [82]. Heatmaps were visualized with the heatmap.2 function in the gplots R package [83]. Clustering was performed by the ward.D2 method using Euclidean distance as a distance measure. Enriched GO terms were identified using Gene Set Enrichment analysis (GSEA) version 4.1.0 [84,85]. GSEA was applied on a ranked gene list based on the log2FC calculated in edgeR. Networks were built in String version 11.5 and visualized in Cytoscape version 3.9.1. [43,44].

### 4.5. Barcode Sequencing and Analysis

For the Bar-Seq experiment the haploid MATa deletion collection was pooled as described by Perez-Samper et al. [86]. Shortly, the deletion collection in 96-well plates was thawed and 3 µL from each well was inoculated in YPD medium containing 200 µg/mL G418. All cultures were grown to stationary phase (OD_600_ > 1.0). 50 µL of each stationary phase culture was pooled and mixed. The pool was divided into 1 mL aliquots and frozen at −80 °C.

Two replicate deletion collection pool aliquots were thawed and pregrown to exponential phase (OD = 0.2) in 2% YPD and an initial sample was taken for each replicate and frozen as a 25% glycerol stock at −80 °C. For each replicate, the exponential phase cultures were diluted in YP in duplicate. One sample was irradiated with 50 Gy PRT or XRT, the other was mock irradiated as a control. After irradiation, the pools were grown for 8 generations in 2% YPD and at end of the growth cycle final samples were taken. Genomic DNA was extracted from all samples using a zymolyase-based protocol. UPTAGs and DNTAGs were amplified by separate PCR reactions using the primers described in Perez-Samper et al. [86]. DNA concentrations were measured with Qubit (Thermo Fisher Scientific, Waltham, MA, USA). Amplified UPTAGs and DNTAGs coming from the same sample were pooled at equal concentrations and sent for paired-end sequencing on an Illumina NextSeq 500 with a 1000× coverage at the Nucleomics Core facility in Leuven.

Sequences were assessed for quality control using FASTQC version 0.11.9 [79]. Barcodes were extracted from the raw reads using cutadapt version 2.3 [87]. Extracted barcodes were matched to the recharacterized deletion barcodes and counted using Barcas version 1.0 [42,88]. A maximum of two inexact matches was allowed. Differential expression analysis was performed using edgeR 3.32.1 [82]. The Log2 fold change in counted barcodes was calculated comparing the irradiated condition to the corresponding non-irradiated condition. Visualization of the results in a heatmap and network and GSEA was achieved similarly as for the RNA-Seq experiment.

### 4.6. Fluorescence Microscopy

Three replicate exponential phase cultures (LAV87 and LAV91) were mock irradiated or irradiated with 50 Gy or 100 Gy PRT or XRT. At indicated timepoints after irradiation, cells were fixed using 4% formaldehyde, washed in PBS and stored at 4 °C in PBS until imaging. In the case of Hsp104 fluorescent foci 50 µM MG-132 (Selleckchem, Houston, TX, USA) was added 2 h before irradiation. YFP fluorescence was imaged using an inverted automated Nikon TiE fluorescence microscope (Nikon, Tokio, Japan) at 60× magnification. In total, 11 fluorescent images were obtained at 0.3 µm intervals along the z-axis. Foci were counted from the fluorescent image using ImageJ. The number of foci positive cells was calculated by dividing the number of fluorescent foci by the total amount of cells as counted on the brightfield image. At least 1500 cells were analyzed per sample.

### 4.7. Cell Cycle Analysis

Three replicate exponential phase cultures of the wild type strain (KV447) were irradiated with 50 Gy or 100 Gy PRT or XRT. Samples were taken at the indicated timepoints, fixated with 70% fresh, ice-cold ethanol and stored at −20 °C. All samples were stained with propidium iodide (PI) for flow cytometry analysis. Briefly, fixated cells were pelleted and washed with sodium citrate buffer (0.035 g citric acid monohydrate (Sigma, St. Louis, MO, USA) and 7.3 g trisodium citrate dihydrate (VWR, Radnor, PA, USA) in 500 mL water). Next, pellets were resuspended in sodium citrate buffer containing 0.25 mg/mL RNase A (Thermo Fisher Scientific, Waltham, MA, USA) and incubated at 50 °C for one hour. Samples were centrifuged and pellets were resuspended in sodium citrate buffer containing 16 μg/mL PI (Sigma-Aldrich, St. Louis, MO, USA). The samples were analyzed with a flow cytometer at 488 nm (FACSverse, BD Biosciences, Franklin Lakes, NJ, USA). A total of 50,000 cells were analyzed per sample.

### 4.8. HAC1 mRNA Splicing Detection

Splicing status of *HAC1* mRNA was detected using RT-PCR across the intron. Therefore, sampling and total RNA extraction was performed similar as described in for RNA-Seq samples. From these samples, cDNA was generated using the QuantiTect Reverse Transcription Kit (Qiagen, Hilden, Germany) following the manufacturers guidelines. This cDNA was used a template for PCR amplification using an intron-flanking primer pair (Appendix A). PCR products were analyzed on a 2% agarose gel.

### 4.9. Spot Assays with MG-132

Three replicate exponential phase cultures (LAV90) were incubated with 50 µM MG-132 (Selleckchem, Houston, TX, USA) for 2 h before mock irradiation or irradiation with 50 Gy or 100 Gy PRT or XRT. After irradiation, 10-fold serial dilutions (start OD = 1) were spotted on YPD containing 50 µM MG-132. Plates were incubated at 30 °C and imaged after 48 h of incubation.

## Figures and Tables

**Figure 1 ijms-23-05493-f001:**
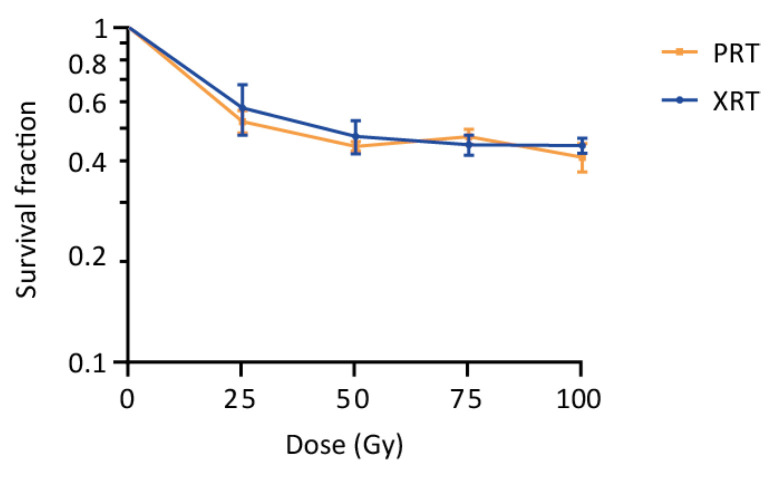
Survival of wild type yeast after proton (PRT) and photon radiotherapy (XRT). Survival of a wild type yeast strain (KV447) after PRT and XRT was determined using plating assays. Survival fractions were calculated and a survival curve was plot on a logarithmic scale. Data are represented as the mean ± SEM for *n* = 3. *p*-values were calculated by multiple t-testing per dose, but no significant differences were detected (*p*-values of 0.66, 0.60, 0.56 and 0.50 were calculated at 25 Gy, 50 Gy, 75 Gy and 100 Gy, respectively).

**Figure 2 ijms-23-05493-f002:**
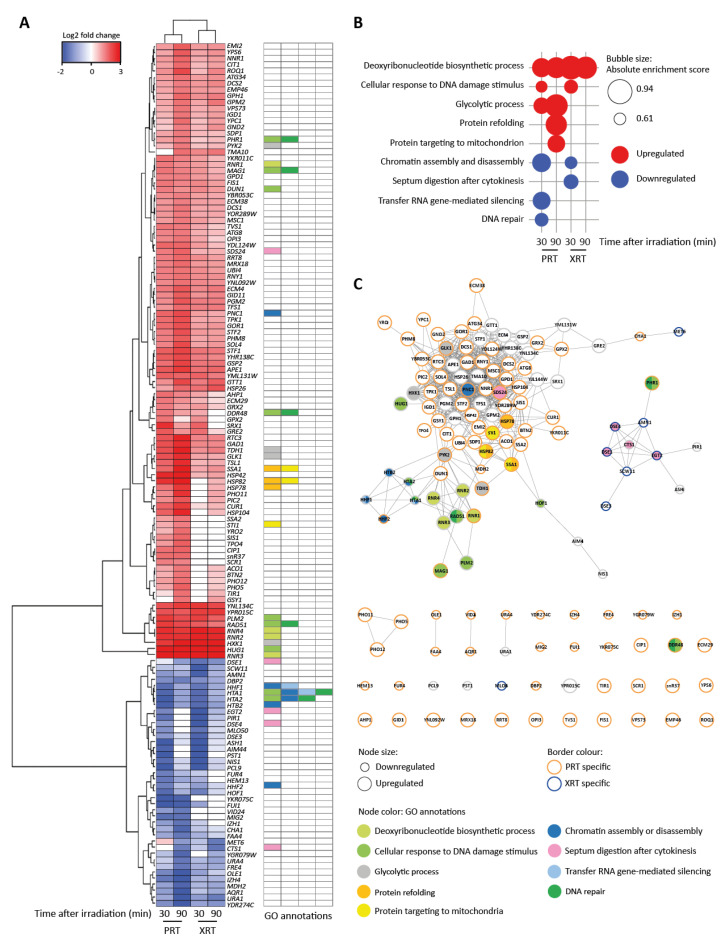
RNA-Seq reveals distinct transcriptional profiles after proton (PRT) and photon radiotherapy (XRT). (**A**) Heatmap for differentially regulated genes. Genes were selected based on two cut-offs: false discovery rate (FDR) < 0.01 and |log2FC| > 1.5 (log2 fold change) in at least one condition. Hierarchical clustering was performed by the Ward.D2 method based on the Euclidian distance. Gene ontology (GO) annotations for enriched GO terms found by gene set enrichment analysis (GSEA) in (**B**) are visualized. (**B**) Enriched GO terms identified using GSEA. GSEA was applied on a ranked gene list based on the log2FC calculated in edgeR. Genes were chosen per condition based on two cut-offs: FDR < 0.01 and log2FC < −1.5 for downregulated genes or log2FC > 1.5 for upregulated genes. For GSEA, the FDR was set at 0.25. (**C**) Interaction network of genes in (**A**). The network was built in String [43] and visualized in Cytoscape [44]. The node size corresponds to either up- or downregulated genes. Nodes with orange borders are genes only found to be deregulated after PRT. Nodes with blue borders are genes only found to be deregulated after XRT. Nodes with grey borders are genes deregulated after both PRT and XRT. Node colors represent the GO categories found in (**B**).

**Figure 3 ijms-23-05493-f003:**
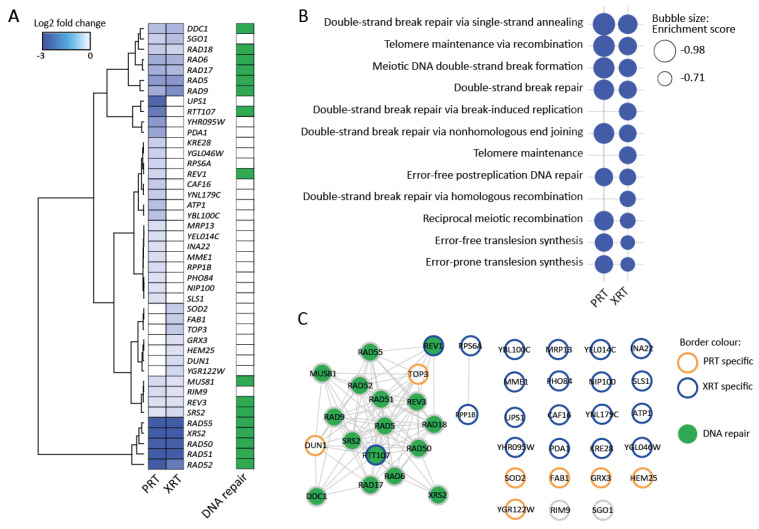
Barcode sequencing (Bar-Seq) experiment identifies DNA repair genes as important for the response to proton (PRT) and photon radiotherapy (XRT). (**A**) Heatmap for sensitive gene deletions. Sensitive gene deletion mutants were selected based on two cut-offs: FDR < 0.05 and log2FC < −0.5 in either the PRT or XRT experiment. Hierarchical clustering was performed by the Ward.D2 method based on the Euclidian distance. Dark green squares indicate genes annotated with the GO term “DNA repair”. (**B**) Enriched GO terms identified using GSEA. GSEA was applied on a ranked gene list based on the log2FC calculated in edgeR. Genes were chosen based on two cut-offs: false discovery rate (FDR) < 0.05 and log2 fold change (log2FC) < −0.5 in either the proton or photon experiment. For GSEA, the FDR was set at 0.25. (**C**) Interaction network of genes in (**A**). The network was built in String [43] and visualized in Cytoscape [44]. Nodes with orange borders are deletions only found to be sensitive after PRT. Nodes with blue borders are deletions only found to be sensitive after XRT. Nodes with grey borders are deletions sensitive to both PRT and XRT. Nodes are colored in green when the gene is involved in DNA repair.

**Figure 4 ijms-23-05493-f004:**
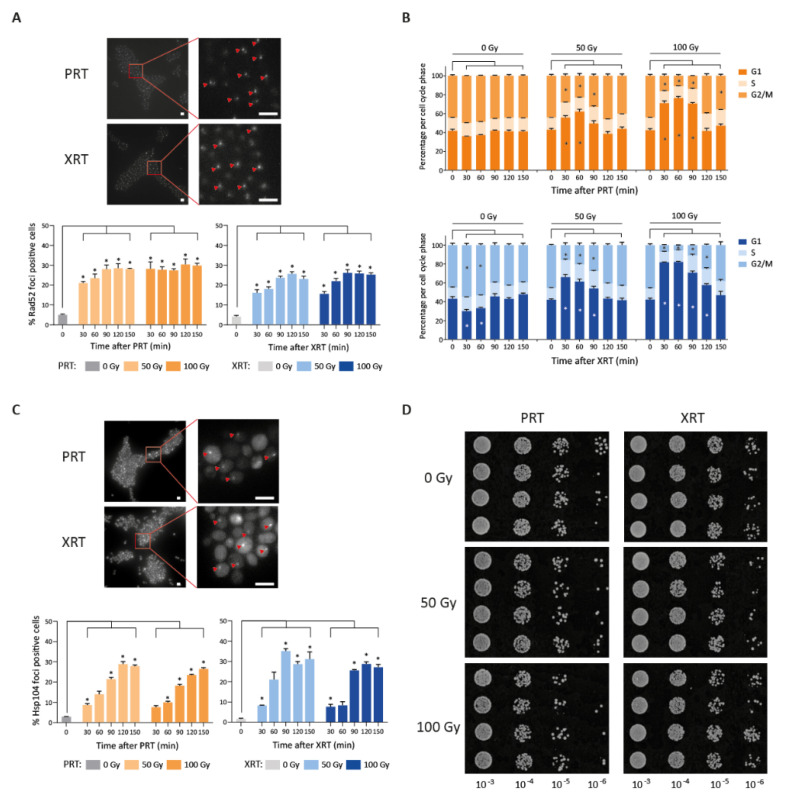
Proton (PRT) and photon (XRT) radiotherapy induce DNA damage and protein damage in *S. cerevisiae*. (**A**) Repair kinetics after photon and proton irradiation. Example image of Rad52 foci 90 min after 100 Gy irradiation (strain LAV87) (top). Scale bars represent 5 µm. The percentage of Rad52 foci positive cells is depicted over time after 50 Gy or 100 Gy of PRT and XRT (bottom). All data are represented as the mean ± SEM for *n* = 3. For each replicate, at least 1500 cells were analyzed per condition. *p*-values were calculated using ANOVA with multiple comparisons test compared to the 0 min control. * *p*-values < 0.05. (**B**) Cell cycle distribution after proton (top) and photon (bottom) irradiation. The percentage of the population in G1, S and G2/M is depicted. *p*-values were calculated using ANOVA with multiple comparisons test compared to the 0 min control. * *p*-values < 0.05. (**C**) Example image of Hsp104 foci 90 min after 100 Gy irradiation (strain LAV91) (top). Scale bars represent 5 µm. The percentage of Hsp104 foci positive cells over time is depicted after 50 Gy and 100 Gy of PRT and XRT (bottom). All data are represented as the mean ± SEM for *n* = 3. For each replicate, at least 1500 cells were analyzed per condition. *p*-values were calculated using ANOVA with multiple comparisons test compared to the 0 min control. * *p*-values < 0.05. (**D**) Spotting assay with proteasome inhibitor MG-132 in combination with 50 Gy or 100 Gy proton or photon radiation. Additionally, 10-fold dilutions of mock irradiated or irradiated cells are spotted on YPD containing 50 µM MG-132. Images were taken after 48 h of incubation at 30 °C.

## Data Availability

The data presented in this study can be made available upon request from the corresponding author.

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
