# Peer review of "An Integrated Approach Reveals DNA Damage and Proteotoxic Stress as Main Effects of Proton Radiation in *S. cerevisiae"

_ijms, 2022, doi:10.3390/ijms23105493_

Round 1
Reviewer 1 Report
Vanderwaeren et al. compared proton radiotherapy (PRT) with x-ray therapy (XRT) in S. cerevisiae and found that both activated the DNA damage response and the protein stress response. However, PRT evoked a stronger activation of proteotoxic stress. Furthermore, the combination of proteasome inhibition and PRT further decreased survival.
Overall, the manuscript is very interesting, well-written, and easy to follow. However, I have a few comments:
Major comments:
The RNAseq analysis showed that genes involved in “deoxyribonucleotide biosynthetic process” were upregulated after both PRT and XRT, and the authors write that “RNA-Seq analysis strongly suggests that both PRT and XRT induce genes involved in DNA repair and hence indicate PRT and XRT evoke a similar DNA damage response”.
Yet, the authors also show that genes involved in DNA repair were downregulated 30 min after PRT. Can the authors elaborate on this?
Can the authors explain why the RNAseq analysis showed that only PRT induced genes involved in protein refolding, yet in the “aggregation experiment”, XRT and PRT induced a similar response?
Where do the authors analyze “Similar to the kinetics observed for DNA damage markers, PRT-induced aggregates show a more persistent pattern compared to XRT”.
Is it based on the bar graph (figure 4C)?. Is it not rather that PRT induces a slower rather than more persistent response? Please clarify.
The authors write: “The results show activation of the UPR for both PRT and XRT, confirming that both PRT and XRT induce proteotoxic stress (Figure S5). Although the extent of UPR activation varied between replicates at different timepoints, on average PRT caused more splicing of HAC1 290 mRNA and thus more ER stress”.
I think the conclusion that PRT casused more ER stress compared to XRT is difficult to make based on the large variation between replicates in the UPR results. Can the authors elaborate?
Minor comments:
The results are written interchangeably in past tense/present tense. Please check
Abstract, line 21: activa-tion vs activation. Please correct
Introduction, line 51-52: Please clarify this sentence: “This hiatus in our understanding has limited to achieve the true potential of PRT but also impeded the development of proton specific therapeutic and combinatorial strategies.“
Results, line 79-80: “Strikingly, although non-significant, the survival fractions obtained for PRT are systematically lower compared to XRT with PRT resulting in on average 1.05 ± 0.07 more cell kill.”.
But that is not the case for 75 Gy? Please clarify.
Line 164: “PRT causes deregulation of much more genes”. Please change “much” to “many”
Figure 4A and C: At what timepoint was the microscopic images (figure 4A) taken?
Reviewer 2 Report
The authors provide detailed scientifically sound experiments to address how proton radiation impacts DNA damage and proteotoxic stress as in S. cerevisiae. A few comments for revisions I have include:
Please provide a bit more detail on why S. Cerevisiae are used as the models. Do they somewhat mimic what is occurring in humans?
Why are so many strains used – is one strain better than other? If so, please explain which strain was used for which experiment
It would be good to validate some key RNA-seq data by western blot for protein expression.
